# Effect of Aluminum Doping Ratios on the Properties of Aluminum-Doped Zinc Oxide Films Deposited by Mist Chemical Vapor Deposition Method Applying for Photocatalysis

**DOI:** 10.3390/nano12020195

**Published:** 2022-01-07

**Authors:** Htet Su Wai, Chaoyang Li

**Affiliations:** School of Systems Engineering, Kochi University of Technology, Kami, Kochi 782-8502, Japan; 246004f@gs.kochi-tech.ac.jp

**Keywords:** aluminum doping ratio, zinc oxide film, photocatalytic property, mist chemical vapor deposition

## Abstract

Aluminum-doped zinc oxide film was deposited on a glass substrate by mist chemical vapor deposition method. The influence of different aluminum doping ratios on the structural and optical properties of zinc oxide film was investigated. The XRD results revealed that the diffraction peak of (101) crystal plane was the dominant peak for the deposited AZO films with the Al doping ratios increasing from 1 wt % to 3 wt %. It was found that the variation of AZO film structures was strongly dependent on the Al/Zn ratios. The intertwined nanosheet structures were obtained when Zn/O ratios were greater than Al/O ratios with the deposition temperature of 400 °C. The optical transmittance of all AZO films was greater than 80% in the visible region. The AZO film deposited with Al doping ratio of 2 wt % showed the highest photocatalytic efficiency between the wavelength of 475 nm and 700 nm, with the high first-order reaction rate of 0.004 min^−1^ under ultraviolet radiation. The mechanism of the AZO film influenced by aluminum doping ratio during mist chemical vapor deposition process was revealed.

## 1. Introduction

In the last decade, the photocatalytic process has been mainly involved in the degradation of chemical pollutants and hazardous industrial waste [1,2]. In 1972, Fujishima et al. observed the unique properties of semiconductor-based photocatalysts on photoelectrochemical water splitting, such as nontoxicity, photosensitivity, and high oxidizing efficiency [3,4,5,6,7]. Since then, many popular metal oxides have been applied as semiconductor photocatalysts, including magnesium oxide, titanium dioxide (TiO_2_), silver oxide, copper (II) oxide and zinc oxide (ZnO), due to their excellent physical and chemical properties [8,9,10,11,12,13]. Among them, ZnO has also emerged as a potential semiconductor photocatalyst due to its promising properties, such as a wide bandgap (3.37 eV) at room temperature, large exciton binding energy (60 meV), super-hydrophilicity, high redox potential, higher surface reactivity, durability, and low cost [14,15,16]. Moreover, it is well established that the main factor in determining the photocatalytic efficiency of ZnO is the photo-excitation process, which involves excitation, diffusion, and surface transfer of photogenerated charged carriers [17,18,19]. However, the faster recombination rate of photogenerated electron-hole pairs can reduce the photocatalytic efficiency of ZnO. It was reported that the recombination rate of electron-hole pairs could be reduced by modifying the surface, introducing the defects or doping with transition metals in ZnO [20,21,22,23,24]. Compared with other methods, aluminum (Al) doping is expected to be a more efficient method of reducing the electron-hole pairs recombination rate of ZnO [25,26,27]. The Al could be doped into ZnO films via different methods, such as radio frequency magnetron sputtering, sol–gel method, chemical vapor deposition (CVD), atomic layer deposition and spray pyrolysis, et al. [28,29,30]. However, there are still issues regarding the controlling of surface morphology, growth direction and the cost [31,32,33]. It was reported that mist CVD had advantages on the film deposition such as high uniformity, high reproducibility, simplicity, easy to control, and low cost [34,35,36,37]. 

In our previous research, we had successfully used the mist CVD method to fabricate the TiO_2_ film [38,39]. In this research, we planned to apply the mist CVD method to synthesize aluminum-doped ZnO (AZO) thin film. The effect of Al doping ratios on the properties of films was investigated, and the photocatalytic activity of AZO films was investigated as well. 

## 2. Experiments

### 2.1. Deposition of AZO Films

AZO thin films (300 nm thick) were fabricated on the glass substrate by the mist CVD method. A solution of mixed precursor was prepared by dissolving the zinc acetate (0.04 mol/L) and aluminum acetylacetonate in water and methanol (90 mL:10 mL). The mist droplets of precursor solution were generated from the solution chamber by ultrasonic transducer at 2.4 MHz, and the droplets transferred to the reaction chamber by nitrogen gas serving as both carrier gas (2.5 L/min) and dilution gas (4.5 L/min). The substrate was set up in the fine channel of the reaction chamber, which was heated and kept at 400 °C during the deposition process. In order to investigate the effect of Al doping ratios on structural and optical properties of AZO films, the doping ratios of Al were set as 1 wt % to 5 wt % for comparison. The deposition conditions are summarized in Table 1.

### 2.2. Photodegradation Process

The photocatalytic measurements of AZO films were carried out in a glass beaker at room temperature. The methyl red (MR) solution with concentration of 1 × 10^−5^ mol/L and volume of 70 mL was prepared, and stirred by wrapping the aluminum foil to prevent the light for 30 min prior to the irradiation. The deposited AZO films with different Al doping ratios were submerged in the MR solution individually and irradiated under an ultraviolet light with a wavelength of 254 nm for 5 h during the measurement process. In order to analyze the degradation rate as the function of reaction time, the irradiated solution was taken out at each 1 h interval.

### 2.3. Characterization

The thicknesses of AZO films were measured by spectroscopic ellipsometry (WVASE32, J.A. Woollam, Co., Inc., Lincoln, CA, USA). The morphologies of AZO films were evaluated by field emission scanning electron microscopy (FE-SEM, SU-8020, Hitachi, Tokyo, Japan) and atomic force microscope (AFM, Nano-R2, Pacific Nanotechnology, Santa Clara, CA, USA). The structural properties of AZO thin films were investigated by grazing incidence X-ray diffraction (GIXRD, ATX-G, Rigaku, Tokyo, Japan). A UV-visible spectrophotometer (U-4100, Hitachi, Tokyo, Japan was used to evaluate the optical properties of AZO films, as well as the absorption spectra of MR solution. All of the measurements were carried out at room temperature.

## 3. Results and Discussion

The AFM images of AZO films deposited with different Al doping ratios by the mist CVD method are shown in Figure 1. The scanning area was 5 × 5 (μm^2^). It was observed that the root mean square (RMS) surface roughness of AZO films decreased from 6.4 nm to 3.7 nm as the Al doping ratio increased from 1 wt % to 2 wt %, then slightly increased from 4.6 nm to 9.7 nm as the Al ratios increased from 3 wt % to 5 wt %. The lowest RMS value of AZO films was obtained at an Al doping ratio of 2 wt %. The RMS values of AZO films are summarized in Table 2.

The GIXRD patterns of AZO films deposited with different Al doping ratios by the mist CVD method are shown in Figure 2. The (101) peak was the dominant peak for the AZO films deposited with Al doping ratios of 1 wt % to 3 wt %, which meant that the growth of ZnO film exhibited the highly preferred orientation along the (101) crystal plane. Three diffraction peaks of (100), (002) and (101) were observed in the AZO films deposited with an Al doping ratio of 1 wt %. The (100) and (002) peaks became weaker when the Al doping ratio was increased to 2 wt %, then disappeared when the Al doping ratios were increased to 3 wt %. No diffraction peaks were observed on AZO films when the Al doping ratios were increased to 4 wt % and 5 wt %, indicating that the obtained films were amorphous. 

The average crystallite size along the (101) crystal plane for AZO films was calculated using Debye–Scherrer’s formula [40]. The crystallite size of AZO films was significantly reduced from 31.82 nm to 13.56 nm when the Al doping ratio was increased from 1 wt % to 3 wt %.

Using the texture coefficient (TC) (hkl) formula [41], the preferred orientation of the (hkl) plane for AZO film could be determined. The TC values of AZO films calculated based on the XRD data from Al ratios of 1 wt % to 3 wt % are shown in Table 3. From the TC value results, it was confirmed that the TC value of (002) peak only existed at an Al doping ratio of 1 wt %. The TC value of the (100) peak was significantly reduced when the Al ratio was increased from 1 wt % to 3 wt %. Compared to the TC value of (002) and (100) orientations, the TC value of (101) peak was the dominated orientation for all three kinds of AZO films, which meant that the preferred growth of AZO films was (101) crystal plane. 

Figure 3 shows the SEM images of AZO films deposited with different Al doping ratios by the mist CVD method. It was clearly observed that the surface morphologies were significantly changed with increased Al doping ratios. The intertwined nanosheet structures were observed only at Al doping ratios of 1 wt % and 2 wt %. The length of the nanosheets were significantly reduced, and the structures were changed into uniform particles as Al doping ratios increased from 3 wt % to 5 wt %. The reason for the structure change might be because of the corresponding variation in the lattice constants of ZnO films influenced by the Al doping ratios. With the increase in Al doping ratios, there was much Al^3+^ (ionic radius 0.53 Å) substituted into ZnO film instead of Zn^2+^ (ionic radius 0.74 Å), which can lead to the structure distortion due to the different ionic radius. 

EDX measurement was carried out in order to evaluate the elemental composition of deposited AZO films with different Al doping ratios by the mist CVD method. The variation of the atomic percentages of Al/Zn, Al/O, Zn/O, and (Al + Zn)/O are shown in Figure 4. It was found that the atomic ratio of Al/Zn was significantly increased as the Al ratios were increased from 1 wt % to 5 wt %, which meant that more Al ions were doped into ZnO, the maximum value was at Al doping ratio of 3 wt %. The atomic ratios of Al/O and (Al + Zn)/O slightly increased; in contrast, that of Zn/O gradually decreased as the Al ratios were increased from 1 wt % to 5 wt %. The atomic ratios of Zn/O were greater than Al/O when the Al doping ratios were at 1 wt % and 2 wt %, which meant that the Zn-O bond was dominated to form ZnO. When the Al doping ratios increased to 3 wt % and furthermore, the atomic ratio of Al/O was greater than that of Zn/O, which indicated that the Al ions served as the oxidizing agent to replace the Zn ion sites incorporated to oxygen ions to form AZO and Al_1−x_O_x_. However, the atomic ratio of (Al + Zn)/O increased and reached a maximum at 2 wt %, then significantly decreased and stayed at nearly level from 3 wt % to 5 wt %, which might be due to the saturation of oxygen ions bonding.

The mechanism of growth direction of AZO films deposited with different Al ratios by the mist CVD method are shown in schematic diagram (Figure 5). The growth of AZO films mainly includes three stages: (a) mist droplets generation, (b) nucleation, and (c) growth process. In the first stage, the mixed precursors of zinc acetate (ZA) and aluminum acetylacetonate (AA) were prepared in the solution chamber and transformed into mist droplets by the ultrasonic transducer. 

In the second stage, the mist droplets including the ZA and AA were transported to the reaction chamber, in which the fine channel substrate was preheated and kept at 400 °C. During the nucleation process, the substrate was heated to 400 °C, which is greater than the decomposition temperature of AA (193 °C) and ZA (237 °C). The decomposed Al ions were much easier to generate than Zn ions to bond with the oxygen ions due to the fact that the decomposition temperature of AA was lower than ZA. Moreover, it was already reported that the bonding mode of AlO_6_ could be confirmed when the deposition temperature was greater than 350 °C during the mist deposition process [42]. Therefore, in this experiment, the Al-O bonds were becoming dominant and forming Al_1−x_O_x_ with the increase in Al doping ratios at the growth temperature of 400 °C, resulting in the suppression of crystal nucleation of ZnO. 

In the final stage, ZnO was formed with the preferential growth in the (101) direction, which might be due to its fastest growth compared to the other growth directions [43]. Therefore, the intertwined nanosheet structures were formed when the Al ratios increased from 1 wt % to 3 wt %.

The optical transmittance of AZO films with different Al doping ratios by the mist CVD method are shown in Figure 6a. The transmittance of all AZO films was more than 80% in the visible region. Moreover, the blue-shift of the absorption edge was observed, which meant that the energy gaps slightly increased when the Al ratios were increased from 1 wt % to 5 wt %.

The optical bandgap variations of AZO film by plotting (αhv)^2^ versus energy of photon (hv) with the different Al ratios are described in Figure 6b. The optical bandgaps of AZO films were calculated by Tauc’s plot equation [44], as in the following;
(αhv)^2^ = A (hv − E_g_)(1)
where A is the constant, h is Planck’s constant, v is the photon frequency and E_g_ is the optical bandgap. The optical bandgaps increased from 3.58 eV to 3.87 eV when the Al doping ratios were increased from 1 wt % to 5 wt %. 

Figure 7a shows the absorption spectra of MR solution for AZO films deposited with different Al doping ratios by the mist CVD method. It was observed that the intensities of absorption spectra decreased between 475 nm and 700 nm when the Al ratios were increased from 1 wt % to 4 wt %, as compared with the original MR solution (black line). The obtained results indicate that photodegradation efficiency was achieved. The lowest absorption intensity was obtained at Al doping ratio of 2 wt %. When the Al ratios increased to 5 wt %, the absorption spectrum was slightly higher than the original solution, which meant that there was no photodegradation efficiency. 

In order to calculate the degradation rate of obtained AZO films, the absorption band at 520 nm was selected, which corresponds to the red color of MR solution. Based on the variations in the intensity of absorption spectra, the rates of photocatalytic degradation are presented in Figure 7b. Langmuir–Hinshelwood (L–H) mode [45] can be used to describe the degradation kinetic of various organic dyes. The adsorption rate may be represented in terms of the coverage ratio of adsorbed reactants on the photocatalyst surface. The rate of reaction, R, can be described as Equation (2) [46].
R = −dC/dt = k_r_θ,(2)
where C is the concentration of the reactant, t is the reaction time, k_r_ is the reaction rate constant, and θ is the coverage ratio of reactants.

According to the adsorption theory, the adsorption capacity and concentration of reactant correspond to the coverage ratio of the reactant. In order to determine the adsorption capacity, the adsorption coefficient of reactant was defined as K. Therefore, the θ can be expressed as Equation (3) [45].
θ = KC/(1 + KC)(3)

Then, Equation (3) can be substituted to Equation (2); the reaction rate, R can be expressed as Equation (4).
R = −dC/dt = k_r_ KC/(1 + KC)(4)

For pseudo first-order reaction, KC, is smaller than 1 [47], then Equation (4) can be expressed as,
R = −dC/dt = k_r_ KC(5)

Equation (5) can be described as per the Integration Law, as follows;
ln (C) − ln (Co) = kt,(6)
ln (C/Co) = kt
where k is the first-order reaction rate constant.

Figure 7b shows the plot of ln (C/Co) as a function of UV irradiation time for photocatalytic degradation of MR solution for AZO films with different Al doping ratios. The value of Co was determined as the original absorbance of MR solution and the value of C was determined as the absorbance of each AZO film with different Al doping ratios. According to the Langmuir–Hinshelwood (L–H) mode, the higher value of first-order rate corresponds to the higher photocatalytic efficiency [48]. 

In order to calculate the first-order reaction rate constant k, the following equation can be rewritten from Equation (6).
k = ln (C/Co)/t(7)

According to Equation (7), the calculated first-order reaction rate increased from 0.003 min^−1^ to 0.004 min^−1^ when the Al ratios was increased from 1 wt % to 2 wt %, then decreased from 0.0005 min^−1^ to 0.0003 min^−1^ when the Al ratios were increased from 3 wt % to 4 wt %. The highest reaction rate was observed from the AZO film with Al doping ratio of 2 wt %. 

## 4. Conclusions

The Al was successfully doped into ZnO film with controllable ratios by the mist CVD method. It was confirmed that the properties of AZO films were significantly influenced by the Al ratios. The highly preferred orientation of AZO films was along the (101) crystal plane when the Al doping ratios were increased from 1 wt % to 3 wt %. The best crystallinity and the lowest surface roughness were obtained at an Al doping ratio of 2 wt %. The crystal structures of AZO were transformed from intertwined nanosheets to uniform particles as the Al doping ratios increased. The growth mechanism revealed that Al-O bonds were enhanced and greater than the Zn-O bond when the Al doping ratios increased at the deposition temperature of 400 °C during the mist CVD process. The transmittance of all AZO films was higher than the 80% in the visible region. The AZO film with Al doping ratio of 2 wt % showed the highest photodegradation efficiency and the highest value of first-order reaction rate of 0.004 min^−1^. The obtained AZO films can be expected to have applications in environmental purification processes.

## Figures and Tables

**Figure 1 nanomaterials-12-00195-f001:**
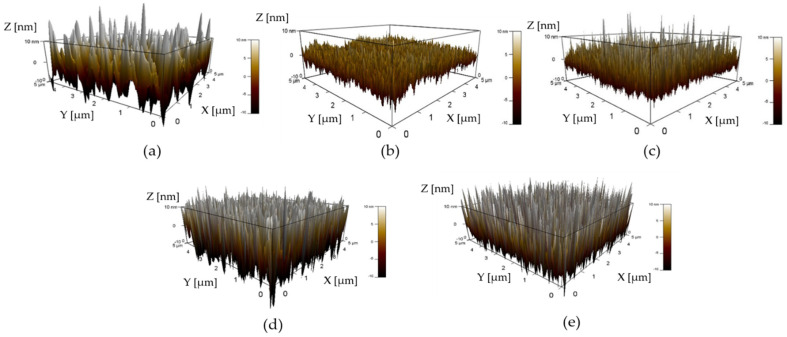
AFM images of AZO films with different Al doping ratios by the mist CVD method: (**a**) 1 wt % Al, (**b**) 2 wt % Al, (**c**) 3 wt % Al, (**d**) 4 wt % Al, and (**e**) 5 wt % Al.

**Figure 2 nanomaterials-12-00195-f002:**
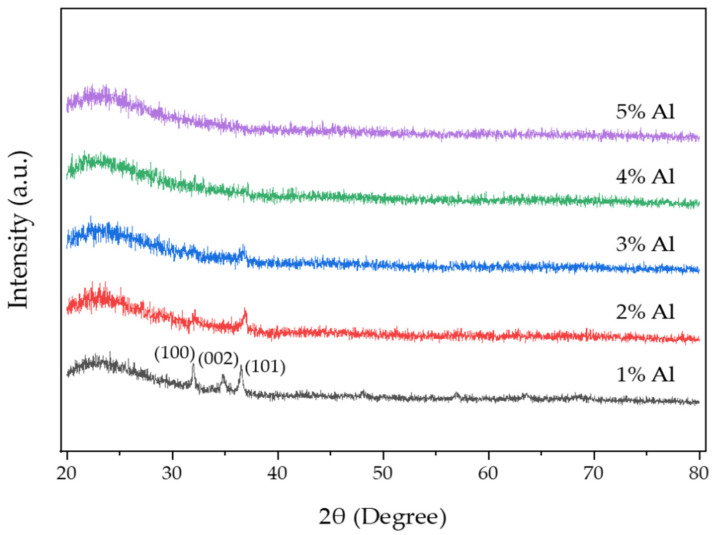
GIXRD patterns of AZO films with different Al ratios by the mist CVD method.

**Figure 3 nanomaterials-12-00195-f003:**
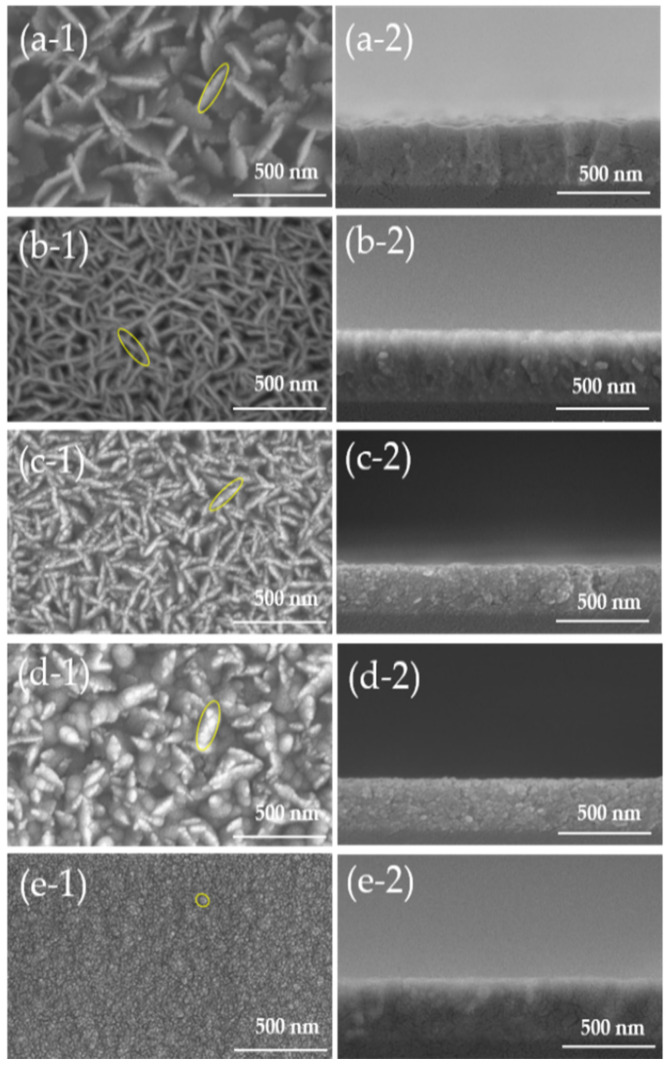
SEM images of AZO films with different Al doping ratios by the mist CVD method. (**a**) 1 wt % Al, (**b**) 2 wt % Al, (**c**) 3 wt % Al, (**d**) 4 wt % Al, and (**e**) 5 wt % Al: (**1**) top view, and (**2**) cross-section view).

**Figure 4 nanomaterials-12-00195-f004:**
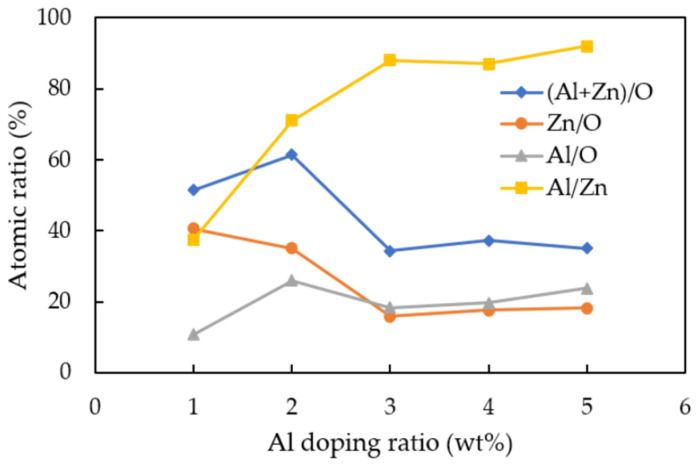
Variations of Al/Zn, Al/O, Zn/O, and (Al + Zn)/O atomic ratios calculated from EDX analysis of AZO films deposited with different Al ratios by the mist CVD method.

**Figure 5 nanomaterials-12-00195-f005:**
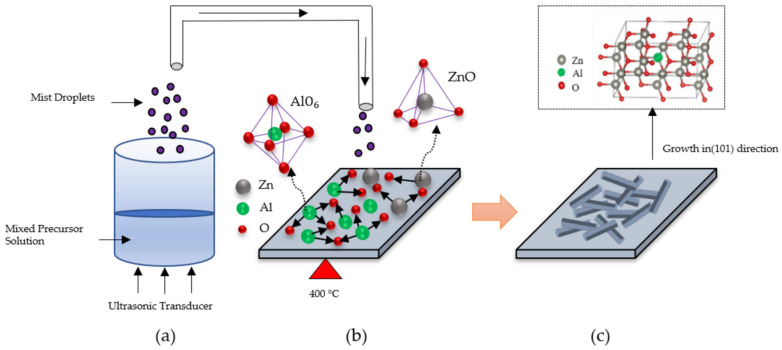
Growth model of AZO films deposited with different Al ratios by the mist CVD method: (**a**) mist droplets generation, (**b**) nucleation, and (**c**) growth process.

**Figure 6 nanomaterials-12-00195-f006:**
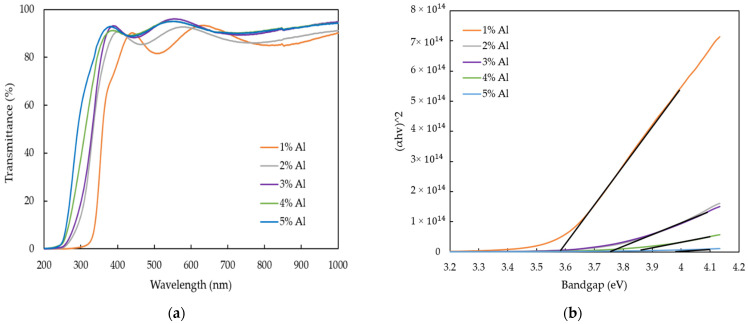
(**a**) Optical transmittance spectra, and (**b**) Variation of bandgap in AZO films deposited with different Al ratios by the mist CVD method.

**Figure 7 nanomaterials-12-00195-f007:**
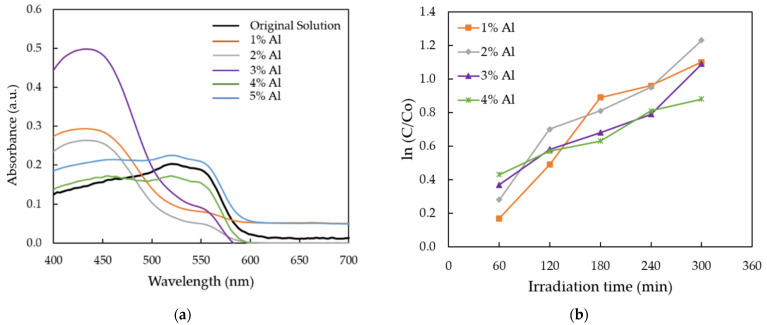
(**a**) The absorption spectra of MR solution for AZO films deposited with different Al doping ratios; and (**b**) plot of ln (C/Co) versus the reaction time.

**Table 1 nanomaterials-12-00195-t001:** Deposition conditions of AZO films by the mist CVD method.

Deposition Parameter	Condition
Solute	Zinc acetate, Aluminum acetylacetonate
Solvent	Methanol, H_2_O
Concentration (mol/L)	0.04
Al doping ratio (wt %)	1, 2, 3, 4, 5
Deposition temperature (°C)	400
Carrier gas, flow rate (L/min)	2.5
Dilution gas, flow rate (L/min)	4.5

**Table 2 nanomaterials-12-00195-t002:** The RMS roughness of AZO films.

Al Doping Ratio (wt %)	RMS Roughness (nm)
1	6.4
2	3.7
3	4.6
4	8.8
5	9.7

**Table 3 nanomaterials-12-00195-t003:** Texture coefficient values of AZO films deposited with different Al ratios.

Al Doping Ratio (wt %)	TC (100)	TC (002)	TC (101)
1	2.785	1.542	3
2	0.126	-	0.732
3	0.117	-	0.654

## Data Availability

All required data are provided within the manuscript.

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
