# Peer review of "Effect of Aluminum Doping Ratios on the Properties of Aluminum-Doped Zinc Oxide Films Deposited by Mist Chemical Vapor Deposition Method Applying for Photocatalysis"

_nanomaterials, 2022, doi:10.3390/nano12020195_

Round 1

Reviewer 1 Report

No ideas

Author Response

Dear reviewer 1,

I would like to appreciate your valuable comments. Please see the attachment in the box.

Reviewer 2 Report

Aluminum doped zinc oxide film was deposited on the glass substrate by chemical vapor deposition method. It was found that the variation of AZO film structures was strongly dependent on the Al/Zn ratios. The optical
transmittance of all AZO films and photocatalytic efficiency were measured. A model of growth was supplied.

I suggest only the following few changes

  • Conclusions: It should be better connected to the introduction specifying how and where  the good photocatalytic activity of material can be applied
  • Results and discussion: I suggest to choose another scale for the Fig. 6b and to add few words of explanation about Tauc plot, although there is a citation, because not all readers know what is a Tauc plot.

Author Response

Dear Reviewer 2,

I would like to appreciate your comments. Please see the attachment in the box.

Reviewer 3 Report

The main criticism consists in incorrect presentation of kinetic data.

  • First of all, absorption spectra of MR solution (Fig 7a) are not understandable.

What moment of photocatalytic experiment do these spectra demonstrate? It should be mentioned.

Unfortunately, fraise (line 205) “When the Al ratios increased to 5 wt%, the absorption spectra was slightly higher than the original solution, which meant that there is no photodegradation efficiency”, means a low precision of photocatalytic experiment. Moreover, in such case how does this 5 wt% Al ratio appears in Fig 7b?

  • ln (C/Co) during photocatalytic degradation of MR should decrease but  in Fig 7b it increases . How it could be? In any case it is mistake.

  • (5) and Eq.(6) should be checked. Line 205 also should be checked – why “ K is the reactant”?

Some additional remarks.

  1. Line 90-91 and Table 2 – the unreal high precision of RMS roughness
  2. Figure 1 – AFM images are nearly the same, it will be better to move these pictures to Supplementary.
  3. Fig 6 and Fig 7 – it should be the same color for the same Al ratios - It is very difficult to follow figures in present state.
  4. Line 221 – the wrong spelling of the Langmuir name.

Author Response

Dear Reviewer 3,

I would like to appreciate your valuable comments. Please see the attachment.

Round 2

Reviewer 3 Report

The text was significantly improved. The content with additional explanations is now clear.

All conclusions are supported by the experimental data.